# Key Role of the Dispersion of Carbon Nanotubes (CNTs) within Epoxy Networks on their Ability to Release

**DOI:** 10.3390/polym12112530

**Published:** 2020-10-29

**Authors:** Maxime Pras, Jean-François Gérard, Luana Golanski, Guilhem Quintard, Jannick Duchet-Rumeau

**Affiliations:** 1UMR CNRS 5223 Ingénierie des Matériaux Polymères, Université de Lyon, INSA Lyon, 20, Avenue Albert Einstein, 69621 Villeurbanne, France; maxime.pras@michelin.com (M.P.); jean-francois.gerard@insa-lyon.fr (J.-F.G.); guilhem.quintard@insa-lyon.fr (G.Q.); 2LITEN, CEA-Grenoble, 17 rue des Martyrs, 38054 Grenoble CEDEX 09, France; luana.golanski@cea.fr

**Keywords:** carbon nanotube, epoxy networks, releasing

## Abstract

Carbon nanotube (CNT)-reinforced nanocomposites represent a unique opportunity in terms of designing advanced materials with mechanical reinforcement and improvements in the electrical and thermal conductivities. However, the toxic effects of these composites on human health have been studied, and very soon, some regulations on CNTs and on composites based on CNTs will be enacted. That is why the release of CNTs during the nanocomposite lifecycle must be controlled. As the releasing depends on the interfacial strength that is stronger between CNTs and polymers compared to CNTs in a CNT agglomerate, two dispersion states—one poorly dispersed versus another well dispersed—are generated and finely described. So, the main aim of this study is to check if the CNT dispersion state has an influence on the CNT releasing potential in the nanocomposite. To well tailor and characterize the CNT dispersion state in the polymer matrix, electronic microscopies (SEM and TEM) and also rheological analysis are carried out to identify whether CNTs are isolated, in bundles, or in agglomerates. When the dispersion state is known and controlled, its influence on the polymerization kinetic and on mechanical properties is discussed. It appears clearly that in the case of a good dispersion state, strong interfaces are generated, linking the isolated nanotubes with the polymer, whereas the CNT cohesion in an agglomerate seems much more weak, and it does not provide any improvement to the polymer matrix. Raman spectroscopy is relevant to analyze the interfacial properties and allows the relationship with the releasing ability of nanocomposites; i.e., CNTs poorly dispersed in the matrix are more readily released when compared to well-dispersed nanocomposites. The tribological tests confirm from released particles granulometry and observations that a CNT dispersion state sufficiently achieved in the nanocomposite avoids single CNT releasing under those solicitations.

## 1. Introduction

Carbon nanotubes (CNTs) always have been very attractive nanofillers because of their remarkable properties. CNTs exhibit an extraordinary large specific surface area depending on diameter combined with an exceptionally high stiffness and electrical conductivity [1]. As a result, CNTs have a real potential for processing the polymer composites and for obtaining mechanical reinforcement but also thermal and electrical conductivities improvement while making the materials more lightweight [2,3,4]. However, the toxicological effects of CNTs on human health are still under study regarding morphological similarities with asbestos and other small fibers [5,6,7]. A release of nanotubes can occur not only during the CNTs production but also during the use of manufactured nanocomposites [8]. Tailoring the releasing of isolated CNT during the nanocomposites lifecycle and especially during the different steps of handling (such as friction, mechanical abrasion, or breakage) is strongly required for a responsible development of these NTC nanofilled polymers materials [9,10]. Up to now, the risk caused by the exposure to CNTs has been hard to evaluate by the lack of data about the impacts on human health and the environment. Precautionary measures have been recommended to protect the workers, users, and the environment. The potential release of CNTs from composite materials as a result of cutting, drilling, sanding, grinding, and UV-light weathering has been investigated [11]. Most of them reported that the particles generated were predominantly micron-sized with protruding CNTs but generally without isolated NTCs [12,13,14]. Only Schlagenhauf et al. showed the release of free-standing individual CNTs from CNT-embedded nanocomposites in addition to abraded particles from the epoxy matrix and agglomerates of CNTs during an abrasion process [15]. Nanocomposites are processed by a wide variety of mixing protocols and include a range of formulations that can impact how CNTs are integrated and encapsulated within the composite. Kohler et al. showed that the likelihood and form of release of nanotubes were determined by the way CNTs are incorporated into the material [16]. In order to understand the toxicological potential of handling CNT-containing composites, characterizing and quantifying the amount of free CNTs versus matrix-bound ones is relevant. A good dispersion state of the nanotubes in the matrix is required to increase the composite performances, but what about the impact of this dispersion state on the CNT-releasing process in the environment? Interactions between a CNT and the surrounding polymer matrix and between CNTs in nanotubes agglomerates are significantly different. Several literature works review the different methods to evaluate and measure the interfacial stress, i.e., IFSS, between a CNT and a polymer matrix and between two CNTs. Even if very different values of IFSS are reported depending on the type of measurement approach, computational simulations versus experimental microdebonding tests, these ones are high are due to the strong physical and chemical bonds at the CNT/polymer interface [17,18,19,20,21]. An average value of 100 MPa can be kept in mind against only 1 MPa for the interfacial stresses existing between two CNTs side by side (such as encountered in CNT agglomerates), which corresponds to the stress necessary to overcome Van der Waals interactions [22,23,24]. As a result, this study consisted of (i) generating different dispersion states of CNT within an epoxy matrix from masterbatch in order to get either individually dispersed CNT or CNT agglomerates, and (ii) finely characterizing those dispersion states thanks to different multiscale approaches in order to well understand the impact of the nanocomposite morphology on its ability to release particles during its lifetime.

## 2. Materials and Methods

### 2.1. Materials

The epoxy prepolymer is the diglycidyl ether of Bisphenol A LY556 from Huntsman Co. (DGEBA,  n® = 0.15, Basel, Switzerland). The hardener chosen, the diaminodiphenylsulfone denoted as DDS from Aldrich (Saint Louis, MO, USA), is added with a stoichiometric ratio aminohydrogen to epoxy equal to 1. This hardener leads to a network with a high *T*g equal to 220 °C after curing 2 h at 150 °C and 3 h at 220 °C. The CNTs used in this study are multiwall carbon nanotubes, MWNTs, provided by Arkema (France) under the form of a DGEBA-based masterbatch (25 wt % MWNT). The CNTs have a mean outer diameter of 12 nm and a mean inner diameter of 5 nm. Their length is from 100 nm to several µm (average length of about 800 nm).

### 2.2. Nanocomposite Processing

In order to get the proper weight fraction of CNT in the nanocomposite, the masterbatch is diluted in the DGEBA prepolymer to get 1 wt % in the DGEBA prepolymer, leading to 0.7 wt % of CNT in the final nanocomposite. Different processing conditions have been applied to generate varying dispersion states, i.e., from large MWNT agglomerates, as in the masterbatch before dispersion, to nearly individually dispersed nanotubes. Two tools with different shearing rates have been compared: (i) a rotary blade Turbotest Rayneri (with a speed of rotation set at 2000 rpm, VMI, Montaigu, France) and (ii) a 3 roll-mill Exakt 80E (Exakt Technologies, Norderstedt, Germany).

### 2.3. Characterization of the Dispersion State

#### 2.3.1. In DGEBA Prepolymer–CNT Suspensions

Investigating the rheological properties of nanofilled suspensions is known to be a powerful method to understand the nanofillers organization into a liquid medium such as DGEBA prepolymer. Frequency sweep tests were performed from 100 to 0.1 rad/s using an AR1000 (TA) rheometer (TA Instruments, Guyancourt, France) with a cone-plate geometry in the linear viscoelastic region of DGEBA/MWNT suspensions.

#### 2.3.2. In Nanocomposites at the Solid State

The morphology of nanocomposites, i.e., cured systems, is observed by Scanning Electronic Microscopy (SEM) using a Philips XL 20 microscope (Eindhoven, Netherlands) and by Transmission Electronic Microscopy (TEM) using a 1200EX Jeol microscope (Germany, France). Specimens for TEM observations have been prepared using ultramicrotomy at room temperature.

### 2.4. Cure Kinetics

#### 2.4.1. Epoxy Conversion Rate and Activation Energy of Reaction

In order to determine if CNTs have an influence on the epoxy conversion rate during the polymerization of epoxy–amine reactive systems, cure kinetics were carried out during 2 h at 150 °C (first step of the curing cycle). The extent of reaction for given times was calculated from the exothermal peak of residual reaction of polycondensation, ∆*Hr* (J/g), compared to the total heat of reaction, ∆*Ht* (J/g). According to the Kissinger method, the activation energy of the condensation reaction can be deduced from differential scanning calorimetry experiments, DSC, at different heating rates, i.e., 2.5, 5, 10, and 15 K.min^−1^, from 30 to 350 °C in order to get a complete reaction. Calorimetric measurements have been performed on the Q20 calorimeter from TA Instruments (France).

#### 2.4.2. Gelation Time Determination

Multifrequency time sweep tests were performed on ARES (TA Instruments France)) with a plate/plate geometry to determine the gel times of the epoxy/amine systems with or without CNTs from the isofrequency crossover point of the tan δ curves during an isothermal curing (at 150 °C).

### 2.5. Nanocomposite Characterization

#### 2.5.1. Mechanical Behavior

The flexural modulus, stress at break, and toughness of nanocomposites were measured and discussed as a function of dispersion states. The flexural properties have been measured at 25 °C on a 2/M device from MTS. Flexural Young’s modulus and stress at break have been determined considering a three-points bending test on machined samples (75 × 15 × 3 mm^3^). The fracture toughness has been evaluated at 25 °C in order to determine critical stress intensity factor K_Ic_ on single edge notched samples (SEN) in three-point bending mode.

#### 2.5.2. Microscopy and Raman Spectroscopy to Analyze the Stress Transfer

Nanocomposites samples were fractured into liquid nitrogen for SEM (Scanning Electronic Microscopy) fracture analysis (Philips XL 20). Plastic deformation at crack tip was also analyzed using TEM (Transmission Electronic Microscopy) on a 1200EX Jeol.

Micro-Raman measurements were realized on the nanocomposites having different dispersion states under compression. The spectra for the CNT were recorded using a Jobin Yvon Raman system (Horiba Jobin Yvon, Palaiseau, France) operating at 514 nm.

### 2.6. Characterization of Particles Release

A tribological process represented in Figure 1 and already described in a previous paper was developed in order to analyze airborne particles that were released by the nanocomposite under mechanical wear [13]. The nanomaterial is disposed in a clean glove box using a light air flow filtered with an H14 HEPA filter, leading to a very low background level of nanoparticles (5–10 cm^−3^). In order to simulate two different types of mechanical wear, a circular brush in steel was used to simulate abrasion and a steel rake was used to simulate scratching. The normal force applied on the sample by the tool is controlled thanks to an analytical balance. A load of 50 g is applied on each sample with a speed of 1500 rotations per minute during 30 s of solicitation followed by 30 s of “rest”. The airborne particles characterization part is composed of a particles collection device on a TEM grid (VTT filtration device), an OPC (optical particle counter) Grimm 1.109, which is a spectrometer giving airborne particles granulometric measurement from 250 nm to 35 µm (optical diameter), and an ELPI (electrical low pressure impactor), which gives the airborne particles granulometric measurement from 7 nm to 10 µm (aerodynamic diameter). In this work, only the ELPI granulometric results will be presented, since the OPC granulometric results were not sensitive enough to provide information at the nanometric scale (lower limit at 250 nm is not appropriate to detect isolated CNT).

## 3. Results and Discussion

### 3.1. Structuration of Carbon Nanotubes

Generating different dispersion states of CNTs in a controlled way remains challenging. Tuning the processing parameters was the chosen pathway to generate extreme morphologies, i.e., aggregated vs. dispersed. The organization of CNT within the epoxy material was studied: (i) before curing, i.e., as the CNT-based masterbatch is dispersed in the DGEBA prepolymer, and (ii) after curing, i.e., in the solid-state nanocomposites.

#### 3.1.1. Structuration of CNT in the DGEBA–CNT Suspensions

The rheology of suspensions is very sensitive to the CNT dispersion state (individually dispersed vs. aggregated) [25,26,27]. In the case of a very good dispersion, the formation of a percolated network can be highlighted from the appearance of a plateau for storage modulus G’ at low frequencies in the linear domain, which is a signature of an elastic response at a given volume fraction. The changes in the G’ dependence with frequency in the low frequencies region, i.e., the exponent of the power law f(w), decreases from 2 (liquid-like) to 0 (gel-like), could be the way to determine clearly the percolation threshold for different volume fractions of CNT. Thus, the rheological signatures of different suspensions (prepared with the same CNT weight percentage) can be compared in terms of dispersion state [25,26,27]. Different dispersion states of 1 wt % CNT in DGEBA prepolymer while varying dispersion tools were studied from 100 to 0.1 rad/s in dynamic mode (Figure 2). As expected, the increase of the mixing time combined with the use of a high shear tool enhances the dispersion efficiency in the prepolymer. The rheological behavior of the initial DGEBA prepolymer displays a Newtonian behavior (storage modulus G’ proportional to ω^2^), whereas a solid-like behavior is evidenced as the prepolymer is filled with 1 wt % of CNT, i.e., G’ becomes independent on the frequency at low frequencies. This evolution of G’ at low frequencies is more pronounced as the 3 roll-mill calander is used. The storage modulus values at the lowest frequency, i.e., at 0.1 rad/s, as well as its dependence at low frequencies according to the power law, G’ ~ ω^∆^, are reported in Figure 2 and Figure 3 as a function of the dispersion conditions. The same conclusions can be made from loss modulus (G″) dependence in frequency.

Adding 1% by weight of MWNT leads to a very pronounced effect on the rheological behavior of the suspension, but it strongly depends on the dispersion state. The rheological signature of the suspension after 5 h of Rayneri dispersion is completely different from the one with the best dispersion state, i.e., obtained using the 3 roll-mill, as G’ increases by almost two orders of magnitude and the exponent of the G’ dependence with frequency is almost divided by four. Hence, the dispersion state of CNT suspension in DGEBA prepolymer, for a given content of CNT, could be quantified using rheological parameters such as the dependence of storage modulus versus frequency in terminal region [28].

#### 3.1.2. Dispersion State Characterization in the Final Nanocomposite

The final morphology at the solid state of nanocomposites results from the freezing of the dispersion state existing on uncured DGEBA/MWNT dispersion and potential changes of the dispersion state during the 3D growing of the epoxy network during cure. Actually, the polymerization process may stabilize the dispersion state obtained before adding the amine hardener, or in the opposite, it may modify the morphology. In order to analyze the CNT dispersion into the thermoset matrix, several methods could be used such as electron microscopies SEM and TEM, which could allow obtaining precise information such as aggregate size distributions, interparticles distances, etc. For the nanocomposites having the poorest dispersion state, SEM microscopy reveals the presence of aggregates and even of agglomerates that have mean sizes around a few tens of micrometers (Figure 4, right). The comparison between nanocomposites having different states of dispersion (Figure 4) shows clearly the efficiency of the 3 roll-mill calander to avoid micron-size agglomerates that are observed after a dispersion of 5 h with a Rayneri rotary blade, which is much less effective.

As reported in a previous paper dedicated to nanoclay-based nanocomposite morphologies, the dispersion needs to be defined at different scales: from large micrometer-size aggregates or agglomerates to the nanoscale, which is the scale of the individual nanofillers [29]. TEM (Transmission Electronic Microscopy) is a powerful method to achieve nanometric-scale characterizations. Dense CNT agglomerates can be seen in Figure 5 on the left corresponding to the morphology of the nanocomposite with the poorest dispersion state, whereas individually dispersed nanotubes obtained with the 3 roll-mill calander are observed in Figure 5 on the right, corresponding to the morphology of the nanocomposite with the best dispersion state.

As those observations allow only a qualitative evaluation of the dispersion, quantitative analyses were performed to assess the meaning of the terms “poorly dispersed” and “well dispersed” CNT-based nanocomposites. An image analysis was performed in order to compute a dispersion degree (DD, defined between 0 and 100; 100 meaning a dispersion only based on individually dispersed CNTs) as well as a micron-size agglomerate percentage µ_agglo%_.
DD= 100 × P_tot_/(P_CNT_ × N_CNT_)(1)
with:P_tot_: Sum of the projected perimeters of the whole particles seen on the micrograph.P_CNT_: Mean projected perimeter of a CNT (504 nm).N_CNT_: Theoretical number of CNTs on the micrographs from the particles’ projected area (N_CNT_ = A_tot_/A_CNT_).A_tot_: Sum of the projected areas of the whole particles seen on the micrograph.A_CNT_: Mean projected area of a CNT (2880 nm^2^).

And
µ_agglo%_ = A_micronic_/A_theoretical_(2)
with:A_micronic_: Sum of the projected areas of the micronic agglomerates seen on the micrograph.A_theoretical_: The theoretical area of the CNT on the micrograph from the volumic filling rate in the nanocomposite.

These discriminating parameters determined for different dispersion conditions are gathered in Table 1. No pre-dispersion means have been used to break the agglomerates before adding the amino hardener.

These discriminating parameters, which are relevant to quantify the dispersion states, highlight that the Rayneri mixer is very relevant to improve the dispersion for short times (DD: +100 % and **µ**_agglo%_: −50 % after 5 h), but it does not enhance drastically the dispersion state for much longer times of mixing. Residual micron-size agglomerates are very difficult to break, and only the use of a 3 roll-mill calander allows decreasing µ_agglo%_ significantly. By the combination of these analysis methods at different scales, extreme dispersion states have been generated and well characterized (with a DD from 5 to 75 and µ_agglo%_ from 70% to 2%).

### 3.2. Impact of Carbon Nanotubes Dispersion State on the Crosslinking Kinetic

As montmorillonite [30,31] or nanosilica [32], carbon nanotubes are well known to have a catalytic effect on the epoxy–amino condensation reactions by the presence of hydroxyl groups on external walls [33,34,35]. As expected, this effect is more pronounced when the volume fraction of CNT increases or when this volume fraction is well dispersed. Let us consider the effects of CNT on the condensation kinetics performed on the reactive system based on DGEBA/DDS/MWNT with 0.7% of MWNT by weight.

#### 3.2.1. Epoxy Conversion Rate

Thanks to isothermal DSC measurements, the extent of reaction and the reaction rate were calculated (Figure 6) as proposed by the research of Xie [35].
(3)$$\alpha =\frac{∆Ht}{∆Ht+∆Hr}$$
with:α is the advance of reaction (epoxy conversion).∆*Ht* is the heat evolved up to a given time (J·g^−1^).∆*Ht +* ∆*Hr* is the total heat of reaction obtained from the isothermal and dynamic scan (J·g^−1^).

After 2 h at 150 °C (first step of the curing cycle), the reaction kinetics for systems prepared with the 3 roll-mill calander pre-dispersion step is faster compared to the unfilled reactive one. On the other hand, the filled mixing without any dispersion step displays a reactivity quite similar to the neat epoxy one and even slightly lower. A clear difference in the cure kinetics is highlighted during the first step of the crosslinking reaction.

The extent of reaction clearly depends of the MWNT dispersion state. After 2 h at 150 °C, α is higher than 80% when the 3 roll-mill was used, and it is equal to 71% in the case of a poor dispersion state. In this case, α is even lower than the extent of reaction of the unfilled reactive system. An agglomerate-based morphology induces a downturn of the reaction kinetics, whereas a good dispersion state has a catalytic effect on the crosslinking reaction.

It could be envisioned that some epoxy prepolymer remains confined in the MWNT agglomerates, leading to a local modification of the epoxy/amine stoichiometric ratio, i.e., impacting reaction kinetics.

#### 3.2.2. Activation Energy of Reaction in the Presence of CNT

The Kissinger method [36] allows determining the activation energy required for a crosslinking reaction [37] from differential scanning calorimetry analyses (DSC) performed at different heating rates.
(4)$$\frac{d(\ln(q/T_{p}²)}{d(1/T_{p})}=-\frac{Ea}{R}$$
with:*q* is the heating rate (K·min^−1^).*R* is the constant of gases.*T_p_* is the temperature at the maximum of the exothermic peak (K).*Ea* is the reaction activation energy (J·mol^−1^).

Figure 7 gives the DSC traces evidencing the exothermic peaks of the DGEBA/DDS systems for the different heating rates.

The polymerization activation energies as a function of dispersion degrees of NTC within the epoxy/amine reactive system are reported in Table 2.

The activation energy of the reactive system is lower in the presence of MWNT whatever the dispersion state compared with the neat epoxy–amine matrix. This decrease of the activation energy is enhanced by a better quality of dispersion (up to 9% of difference between the dispersion obtained with the 3 roll-mill calander and the system without pre-dispersion).

#### 3.2.3. Gelation Time

Gelation time determination provides information about the catalytic effect of carbon nanoparticles and their influence on the changes in the physical behavior of the reactive system. The formation of a chemical three-dimensional network leads to the appearance of an elastic response during polymerization. The gelation time corresponds to the transition from a liquid state to a gel, which was experimentally measured by chemio-rheological analyses from the crossover of the loss factor tan δ vs. time during a multi frequency time sweep. Gel times are measured at 150 °C on the reactive systems based on DGEBA/DDS filled with 0.7% of MWNT by weight tuned as a function of the dispersion state.

Figure 8 shows that the finer the dispersion, the shorter the gel time. On the other hand, an agglomerated state leads to a longer gel time than that measured on the unfilled reactive system. This observation can be explained once again by the confinement of a small quantity of prepolymer within MWNT agglomerates that does not participate in the growing of the network, i.e., local changes of the stoichiometric ratio are generated in and close to the CNT aggregates.

As a conclusion, it appears that the MWNT dispersion state has a pronounced influence on the epoxy–amine system polymerization. The catalytic effect of CNT is promoted by a good dispersion state, whereas a poor dispersion induces a downturn of the crosslinking reaction.

### 3.3. Impact of CNT Dispersion on the Mechanical Behavior of Nanocomposites

The nanocomposites having a well-defined dispersion state were selected to study their mechanical behavior. Indirect and direct analyses were carried out in order to understand the stress transfer from matrix to CNT and between CNT in the aggregates or agglomerates.

#### 3.3.1. Macroscopic Analyses

Three-point bending tests carried out on nanocomposites are reported in Figure 9 and Figure 10. As reported in the literature [12,20,22,38], the mechanical properties are very sensitive to the dispersion state. Thus, a poor CNT dispersion state (DD = 4.9) leads to a decrease of more than 20% for the Young’s modulus, whereas the best dispersion state (DD = 74.5) leads to an increase of 10% compared to the neat epoxy network one. The stress at break is also very sensitive to the dispersion state, since a decrease of 40% with the poorest dispersion state and an increase of 35% in the case of a good dispersion state are reported.

The increase of the Young’s modulus and of the flexural stress at break associated to a good dispersion state of CNT can be explained by an efficient stress transfer at the CNT/polymer interface as the CNT surfaces are fully in interaction with the epoxy network. On the other hand, the significant decreases of these properties as the polymer matrix is filled with micronic MWNT agglomerates can be explained by the poor cohesion of agglomerates that is only governed by the Van der Waals interactions between MWNTs. The fracture toughness of nanocomposites (Figure 11) is also very sensitive to the dispersion state. K_IC_ decreases by 50% with a poor dispersion state and increases up to 22% for the best dispersion state (reached with 3 roll-mill) compared to the neat epoxy network. A good dispersion state is required to gain benefit from multiwall carbon nanotubes on the mechanical properties of nanocomposites. Otherwise, an agglomerated morphology can strongly reduce the mechanical properties of a neat epoxy network. The decrease of mechanical properties of nanocomposites compared to the neat network ones can be directly linked to the non-polymerized DGEBA prepolymer confined in the CNT agglomerates and/or to non-stoichiometric zones, as reported previously from reaction kinetics analyses.

This reinforcing effect of CNT can be explained by the “crack-front pinning” process at the crack tip [38]. In such a process, the crack-front length increases as the crack meets rigid inclusions. As a consequence, this effect is magnified as the number of obstacles increases, i.e., in the case of a good dispersion state. If the MWNTs are agglomerated, the number of objects decreases, and their cohesion being weak (Van der Waals interactions), secondary cracks are generated.

As a conclusion, it appears that as a small amount of MWNT is well dispersed in the matrix, the mechanical properties are significantly improved compared to the unfilled polymer, but they can be seriously affected if the dispersion state is not good enough. In the first case, these are the CNT/polymer interactions, which govern the behavior of well-dispersed CNT-based nanocomposites, whereas in the second case, this is the level of interaction between CNT that governs the behavior of poorly dispersed CNT-based nanocomposites.

#### 3.3.2. Analysis of Interfacial Interactions CNT/CNT vs. CNT/Matrix

Interfacial interactions with CNT can be also analyzed more locally from scanning electronic microscopy (SEM) and transmission electronic microscopy (TEM) on fracture surfaces.

Epoxy networks typically exhibit a brittle fracture, which can be characterized by a resulting mirror-like fracture surface. The presence of MWNT aggregated under the form of agglomerates does not modify so much the fracture surface of poorly dispersed CNT-based nanocomposites (Figure 12-left). On the other hand, the toughening effect of well-dispersed MWNT is clearly shown in SEM picture (Figure 12, right). During failure, carbon nanotubes that are well wetted by the epoxy matrix induce huge plastic deformation. As a consequence, the fracture toughness increases. The roughness of the fracture surface and the presence of crack pinning are signatures of good interactions between MWNT and epoxy matrix.

In order to go further into the investigations of fracture mechanisms, TEM micrographs were realized on crack ends for the different materials (Figure 13).

Individually dispersed CNTs embedded in polymer matrix are clearly observed on the fracture surface of a nanocomposite with a good dispersion state, which is the signature of a good wetting (Figure 13, above) and a strong interfacial adhesion. The rough fracture profile confirms the crack-front pinning effect, leading to the increase of the fracture toughness of the nanocomposite. On the other side, for the poorly dispersed CNT-based nanocomposite (Figure 13, below), MWNTs are pulled out from the polymer, showing that they are not embedded by the polymer matrix. The fracture surface is smooth and regular, even if the crack front meets a MWNT agglomerate. Its propagation front does not seem be affected and stays straight (no toughening effect).

From those electron microscopy observations, the interactions between CNT/CNT (in agglomerates) and CNT/polymer appear clearly different; i.e., isolated CNT will require more energy to be extracted from the surrounding polymer than ones confined in an agglomerate.

Another experimental method to measure the level of the interaction between a CNT and the polymer matrix is micro-Raman spectroscopy. Previous works showed that the Raman spectra of carbon fibers were sensitive to the stresses induced in the fibers along their axis [39,40]. Thus, the stress transfer between the polymer matrix and CNT can be assessed by considering the shift of the G’ band (2700 cm^−1^) in the nanocomposite submitted to a compression or traction load (Figure 14).

Nanocomposites with very different dispersion states were placed under compression (1% and 2% of strain), and the G’ band location was followed as a function of the MWNT dispersion state (as shown in Figure 15).

In case of a well-dispersed CNT-based nanocomposite under compression, a slight shift of the G’ band is observed, which means that the MWNTs sustain compression thanks to a good stress transfer. In the case of a nanocomposite with poorly dispersed CNT, no shift of the G’ band is observed whatever the strain applied to the nanocomposite. MWNTs in agglomerates probably tend to be reorganized under the matrix deformation, and only a few CNTs sustain compression. Nevertheless, let us note that the G’ band location in both extreme dispersions is very different even without applying compression. This can be explained by the polymer shrinking during the crosslinking, applying an additional compression on individually dispersed MWNT.

### 3.4. Impact of CNT Dispersion on the Potential Release Ability

Let us consider now the effect of the dispersion of fillers on the potential release ability during the nanocomposite life. The analysis of released particles investigated after simulating the wear undertaken by the material by using two tools: either a brush or a rake is reported under the form of size distribution in Figure 16 and Figure 17. A releasing of ultrathin particles can be noted with a maximum diameter lower than 100 nm. The tool used has a non-negligible influence on the releasing. The rake allows releasing a higher concentration of nanoparticles (22,600 cm^−3^) with a much lower diameter (7–28 nm) in comparison with the brush (6000 cm^−3^). This may be explained by the friction coefficient that is higher with the rake than with the brush, applying a higher equivalent tensile stress from the Von Mises yield criterion. The hardness of the material has also an influence on the stress applied by the tool on the sample and consequently on the releasing. The stiffness and the hardness of the glassy epoxy networks is prone to generate more particles with lower size (Wohlleben et al. [41]).

At the end, the dispersion state is a significant parameter on the granulometry of released particles. The nanocomposite with a bad dispersion releases much more nanofillers (between 50 and 70% fillers with the lowest diameters) than well-dispersed or unfilled ones that display the same granulometric profile. The aggregates or agglomerates behave as defects favoring the crack propagation and initiation within the nanocomposite. Under the crack effect, the particles are released out of aggregates, where the NTC are embedded into a confined DGEBA prepolymer. Only aggregated particles with protruding fibers are observed in TEM picture reported in Figure 18, whereas free-standing CNTs were not observed.

## 4. Conclusions

This paper aims at generating different dispersion states but in a controlled way of carbon nanotubes into an epoxy–amine network. Two extreme morphologies are designed by varying the dispersion tools, one with micron-size agglomerates of aggregated carbon nanotubes characterized by a high micron-size agglomerates percentage and another one with individual carbon nanotubes embedded into epoxy matrix, which is characterized by a high dispersion degree (DD). The dispersion state of carbon nanotubes has an outstanding effect on the network build up: the gelation is catalyzed and the conversion is enhanced when the carbon nanotubes are well dispersed. In the other case, epoxy prepolymer can be confined into the carbon nanotubes aggregates locally modifying the epoxy–amine stoichiometry preventing or delaying a total conversion. A well-dispersed morphology is the key architecture to reinforce the mechanical resistance of the nanocomposite. Well embedded in the surrounding polymer, the carbon nanotubes ensure a good load transfer toward the matrix. On the contrary, a poor dispersion state with micron-size CNT agglomerates has quite no influence on the crack direction. If the crack front agglomerates on its path, it just goes through considering the weak agglomerate cohesion. There is no strong load transfer between CNT in agglomerate, and the mechanical properties of composites containing micronic agglomerates of MWNT can be much lower than the initial properties of the unfilled polymer. After showing the huge influence of the NTC dispersion on the polymerization mechanisms at a macromolecular scale and on the macroscopic behavior of nanocomposites, these works established a link between the nanocomposites morphology and the potential release of carbon nanotubes during the nanocomposite life. We checked experimentally that less energy is required to extract a CNT from an agglomerate than to extract a CNT individually dispersed in the polymer matrix.

## Figures and Tables

**Figure 1 polymers-12-02530-f001:**
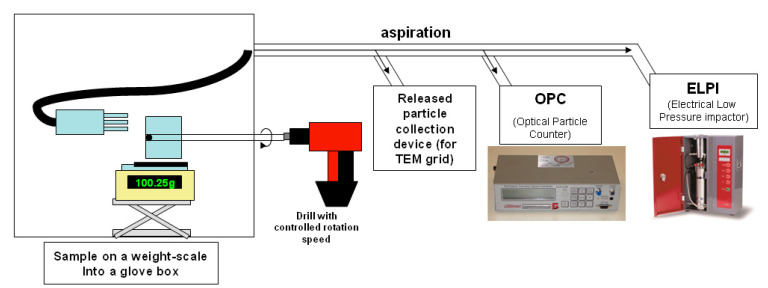
Representation of the wear simulation device in order to analyze airborne released particles.

**Figure 2 polymers-12-02530-f002:**
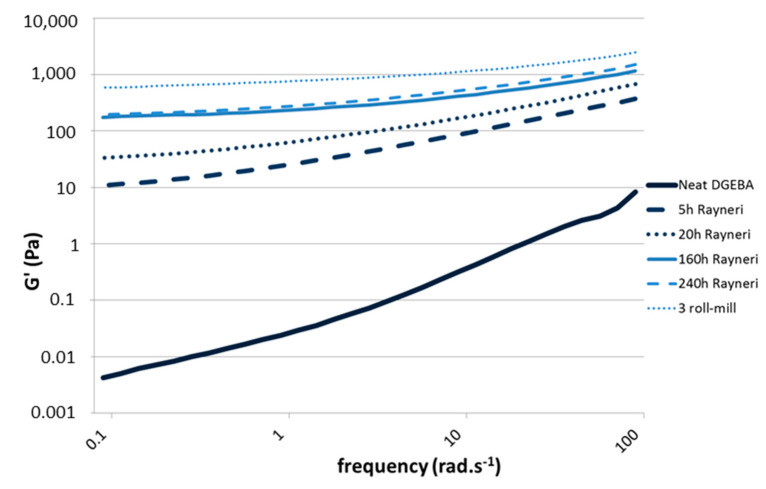
Evolution of storage modulus (G’) as a function of frequency, 0.1% of dynamic strain at 25 °C, for diglycidyl ether of Bisphenol A (DGEBA)/multiwall carbon nanotube (MWNT) suspensions (1 wt %) with different mixing conditions.

**Figure 3 polymers-12-02530-f003:**
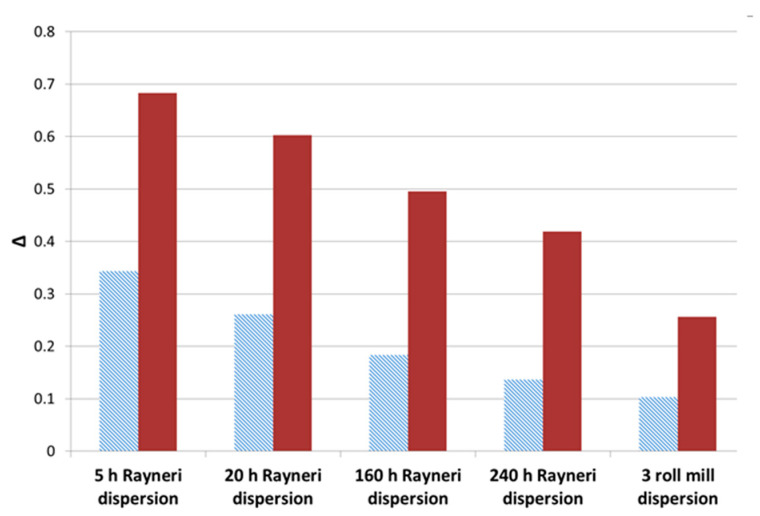
∆ exponent from the power law of G’ (hatched light) and G’’ (full dark) with the frequency (from 0.1 to 1 rad/s) at 25 °C for DGEBA/MWNT suspensions (1 wt %) with different mixing conditions.

**Figure 4 polymers-12-02530-f004:**
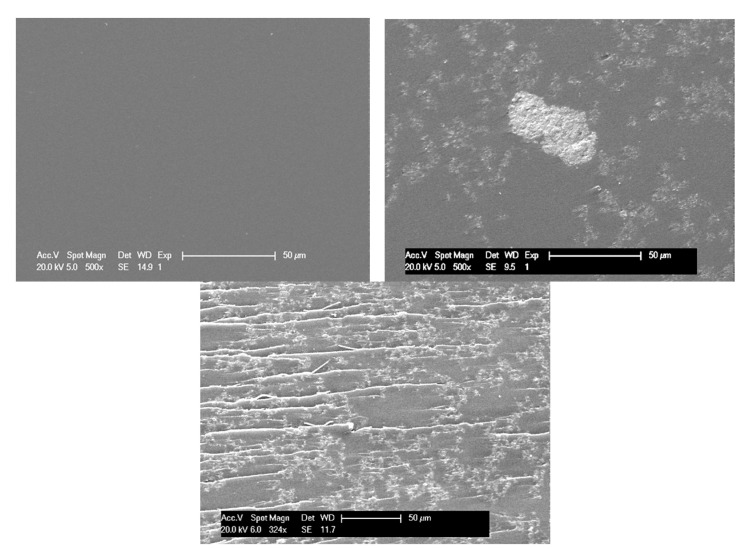
SEM micrograph of unfilled epoxy network (**left**), epoxy/diaminodiphenylsulfone (DDS)/MWNT nanocomposite (0.7 wt %) after a 5 h dispersion with Rayneri (**right**) and epoxy/DDS/MWNT nanocomposite (0.7 wt %) after a dispersion with 3 roll-mill calander (**below**).

**Figure 5 polymers-12-02530-f005:**
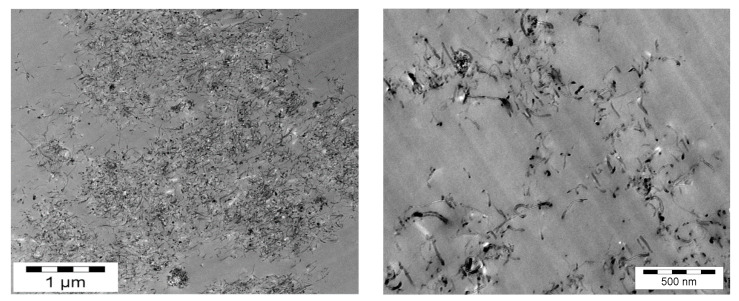
TEM micrographs of epoxy/DDS/MWNT nanocomposites, (0.7 wt %) after a 5 h dispersion with Rayneri (**left**) and after a dispersion with the 3 roll-mill calander (**right**).

**Figure 6 polymers-12-02530-f006:**
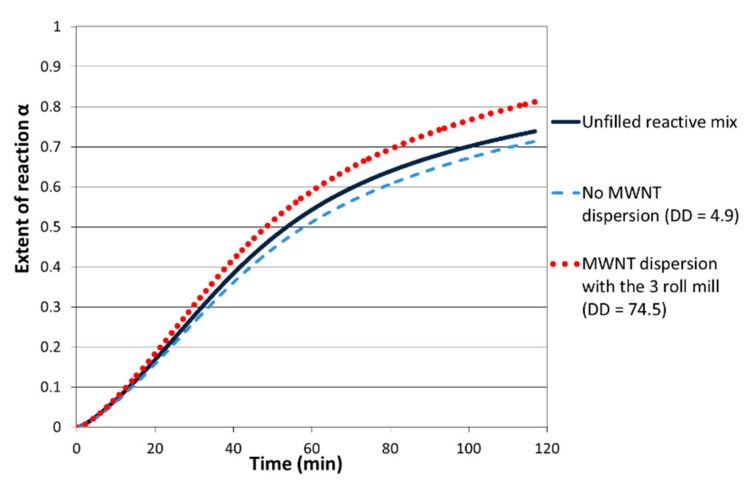
Extent of reaction (α) vs. time for a DGEBA DDS reactive system: without MWNT (continuous line) and filled with MWNT (0.7% in weight) with a different dispersion state (slashed line for the poor dispersion state and dotted line for the good dispersion state) at 150 °C.

**Figure 7 polymers-12-02530-f007:**
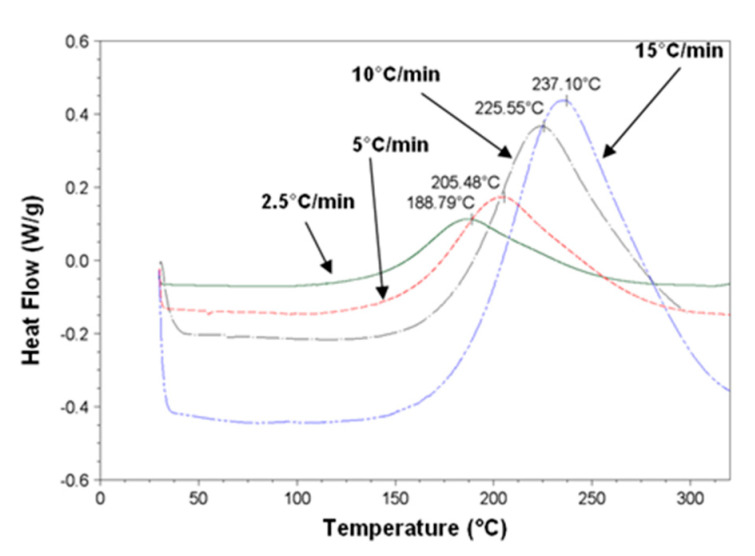
DSC analyses on reactive epoxy/amine (E/A) systems for different heating rates.

**Figure 8 polymers-12-02530-f008:**
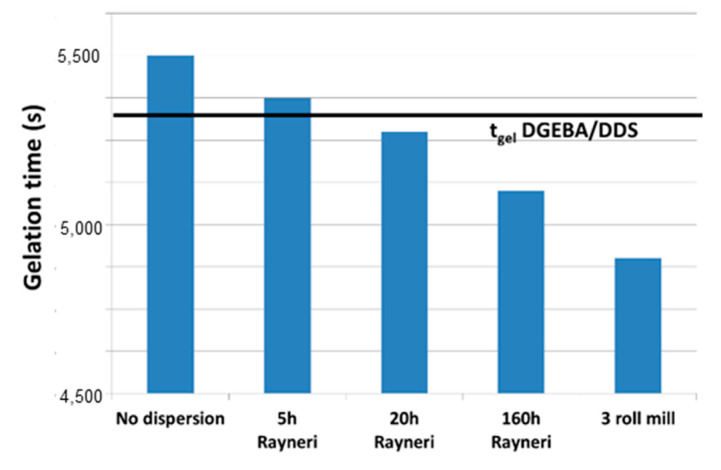
Gel time evolution as a function of the mixing conditions for the reactive system epoxy/amine/MWNT (0.7 wt %) at 150 °C.

**Figure 9 polymers-12-02530-f009:**
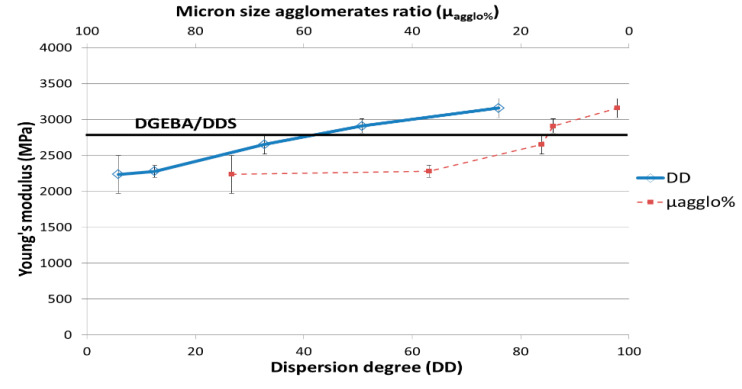
Young’s modulus as a function of MWNT dispersion degree in the DGEBA/DDS/MWNT nanocomposites (0.7 wt %).

**Figure 10 polymers-12-02530-f010:**
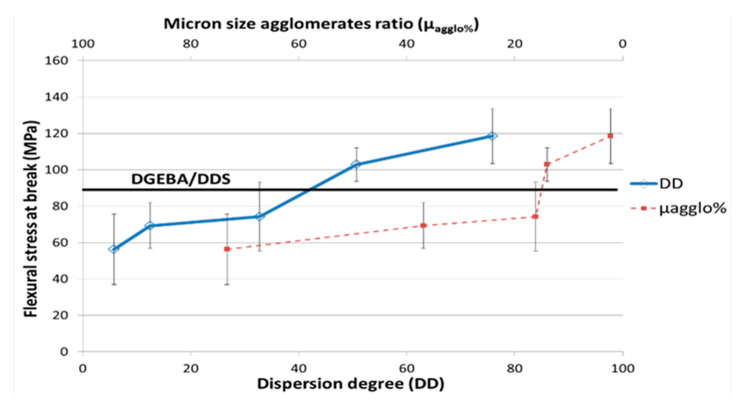
Flexural stress at break as a function of MWNT dispersion degree in the DGEBA/DDS/MWNT nanocomposites (0.7 wt %).

**Figure 11 polymers-12-02530-f011:**
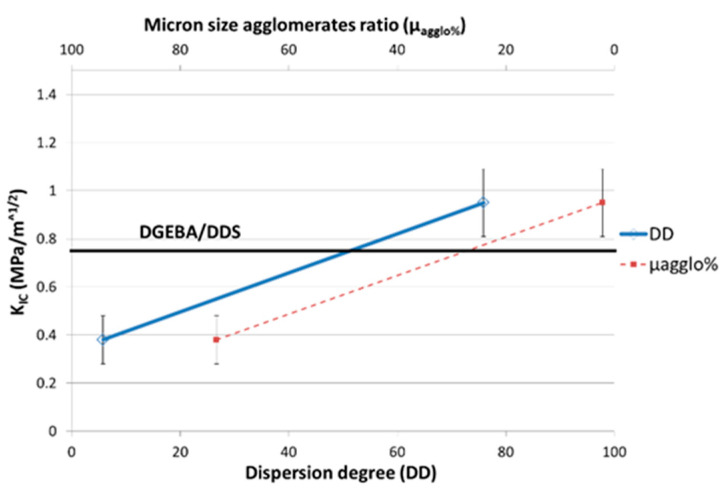
Fracture toughness (critical stress intensity factor K_IC_) as a function of the MWNT dispersion degree in the DGEBA/DDS/MWNT nanocomposites (0.7 wt %).

**Figure 12 polymers-12-02530-f012:**
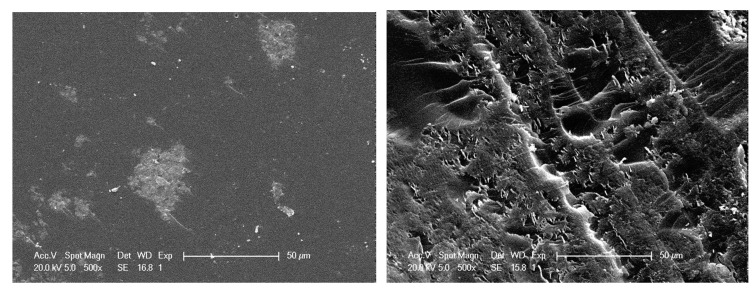
SEM analysis on fractured surfaces of DGEBA/DDS/MWNT nanocomposites (0.7 wt %) with a poor dispersion state, DD = 4.9 (**left**), and with a good dispersion state, DD = 74.5 (**right**).

**Figure 13 polymers-12-02530-f013:**
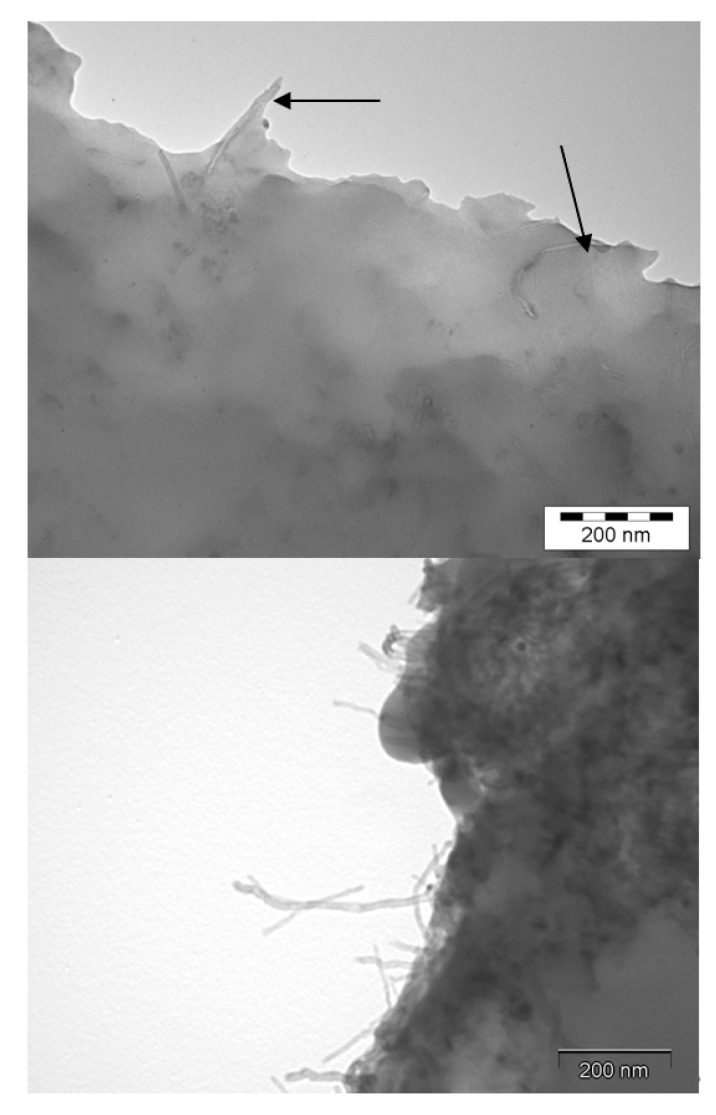
TEM micrographs on the fracture surfaces of DGEBA/DDS/MWNT nanocomposites (0.7 wt %) (**above**) After dispersion using a 3 roll-mill calander, (**below**) without pre-dispersion.

**Figure 14 polymers-12-02530-f014:**
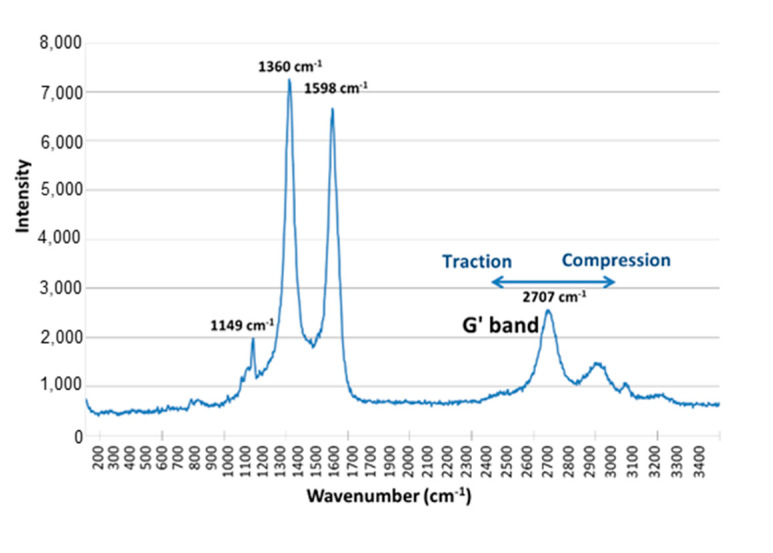
Raman spectrum of a MWNT-based agglomerate in DGEBA/DDS/MWNT nanocomposite (0.7 wt %, DD = 4.9).

**Figure 15 polymers-12-02530-f015:**
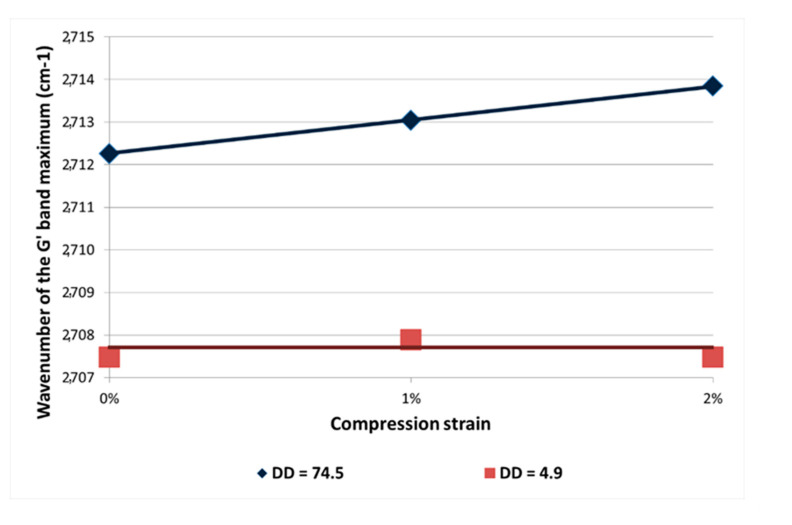
G’ band shift as a function of the compression strain of DGEBA/DDS/MWNT nanocomposite (0.7 wt %) and of two different dispersion states (DD = 4.9 and DD = 74.5).

**Figure 16 polymers-12-02530-f016:**
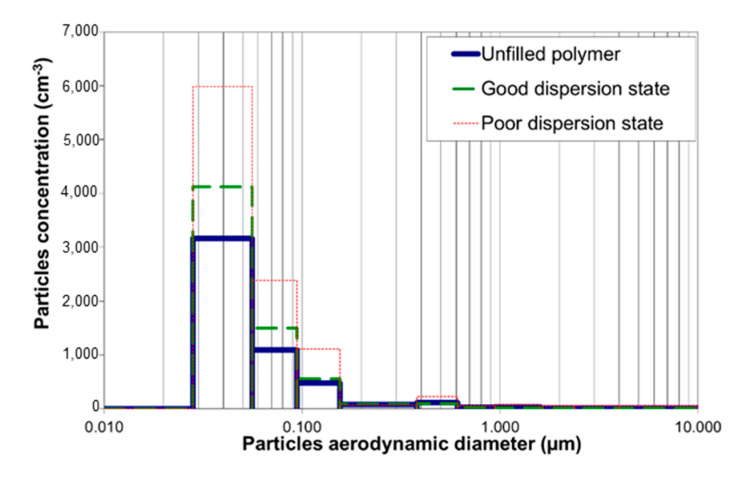
Granulometric distribution of released particles during the use of the steel brush on DGEBA/DDS samples (electrical low pressure impactor (ELPI) data

**Figure 17 polymers-12-02530-f017:**
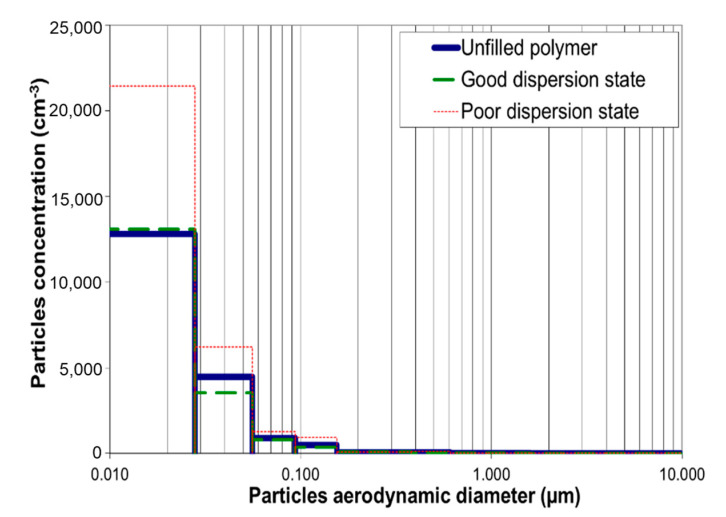
Granulometric distribution of released particles during the use of the steel rake on DGEBA/DDS samples (ELPI data).

**Figure 18 polymers-12-02530-f018:**
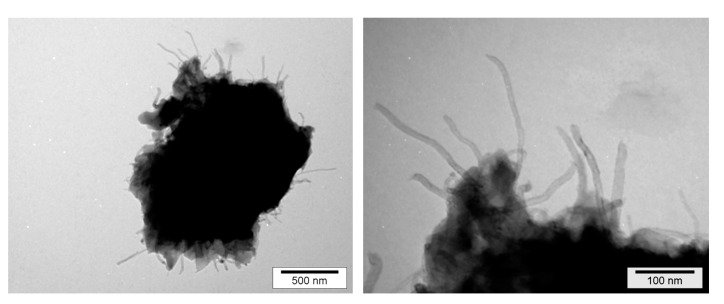
Released particle after wear of DGEBA/DDS/MWNT (0.7 wt %) nanocomposite with a bad dispersion, DD = 4.9.

**Table 1 polymers-12-02530-t001:** Dispersion degree (DD) and micron-size agglomerates percentage (µagglo%) for the different DGEBA/DDS/MWNT nanocomposites (0.7 wt %) processed in different conditions. (standard deviation in brackets).

	No Pre-Dispersion	5 h Rayneri	20 h Rayneri	160 h Rayneri	240 h Rayneri	3 Roll-Mill
**DD**	**4.9** (1.4)	**10.6** (3)	**28** (10.3)	**42** (18.7)	**44.9** (13.1)	**74.5** (15.8)
**µ** _agglo%_	**73.3** (32.3)	**36.9** (20.8)	**16.1** (11.5)	**14** (16.9)	**13.4** (2.3)	**2.2** (3.6)

**Table 2 polymers-12-02530-t002:** Polymerization activation energies of the DGEBA/DDS reactive systems filled with MWNT (0.7% by weight) for different dispersion degrees (DD).

DD	Ea (kJ·mol^−1^)
Neat	71.9
4.9	71.1
10.3	70.7
28.35	69.4
47.6	69.4
74	65.9

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
