# Peer review of "Key Role of the Dispersion of Carbon Nanotubes (CNTs) within Epoxy Networks on their Ability to Release"

_polymers, 2020, doi:10.3390/polym12112530_

Round 1

Reviewer 1 Report

Very important article from the point of view of human health, interesting and carefully prepared. A wide range of research, extensively described with an in-depth analysis of the research results. My only serious comment to the article is the lack of literature from the last 3 years (there is only one article from 2017, then 2 from 2012 and 1 from 2011). The fact that the topic of toxicity was very actual in the early 21st century (2000-2010 years) but it is not possible that no articles have been appeared on that subject in recent years. The authors could search for newest ones and supplement the introduction with the latest results in this field of research.

The small comments to the authors are as follows:

On page 7 lines 221 and 223 there are no literature sources (error - reference source not found), the same page 8 row 264, page 13 rows 384 and 386 and page 14 row 401.

Additionally, on page 9 line 288 there is a reference number [68], which does not exist in the bibliography.

The last remark concerns the drawing no. 13 - the bottom drawing, which appeared twice in the text.

Despite these criticisms, I believe that this is a very valuable article that, after updating the literature and correcting the mistakes mentioned, is worth publishing in the Polymers Journal.

Author Response

Dear reviewer

Thank you for your comments and suggestions.

We added references related to more recent works

Francis et al 2018, Kohler et al 2008, Nowacka 2013, Petersen 2011, Petersen 2014

We added a reference on page 7 line 221 & 223 on the rheology of nanofilled polymers.

We corrected the number of references and removed the duplicated figure 13.

Hoping it matches your expectation

Reviewer 2 Report

This paper presented an investigation of the release and dispersion of CNTs in polymer matrix composites. The authors first reviewed the state of the art. Then detailed experimental procedures were presented. The authors reported adequate results. However, some of the results were not presented clearly. The reviewer suggests a revision before publishing this paper. Please see detailed comments listed below:

  1. Please check the figures in Page 14. It seems the second figure was duplicated in this page.
  2. Many approaches have been reported to increase the CNT dispersion in polymers. One type of approach is through manufacturing procedures, such as ultrasonication. The other method is through the materials preparation, such as CNT functionalization. Please provide a more comprehensive review of the state of the art in the introduction.

Author Response

Dear reviewer,

Thank you for your comments.

The state of the art is focused on the release of carbon nanotubes from nanocomposite during the use of manufactured nanocomposites. This work is focused on the ability of nanocomposites to release carbon nanotubes as a function of their dispersion state. Using ultrasounds is not possible because this works uses epoxy/CNT masterbatch providing from supplier which is more relevant about safety of workers. epoxy/CNT Masterbatch is viscous and require to be introduced into extruder and diluted with additional epoxy. For the same reasons, functionalization of carbon nanotubes can not be a strategy since carbon nanotubes can not be supplied alone but within epoxy based masterbatch.

We removed the Figure 13 duplicated.

Best regards.
